# Increased Herpesvirus Entry Mediator Expression on Circulating Monocytes and Subsets Predicts Poor Outcomes in Pancreatic Ductal Adenocarcinoma Patients

**DOI:** 10.3390/ijms26072875

**Published:** 2025-03-21

**Authors:** Isabelle Kuchenreuther, Finn-Niklas Clausen, Johanne Mazurie, Sushmita Paul, Franziska Czubayko, Anke Mittelstädt, Ann-Kathrin Koch, Alara Karabiber, Frederik J. Hansen, Lisa-Sophie Arnold, Nadine Weisel, Susanne Merkel, Maximilian Brunner, Christian Krautz, Julio Vera, Robert Grützmann, Georg F. Weber, Paul David

**Affiliations:** 1Department of Surgery, University Hospital Erlangen, 91054 Erlangen, Germany; isi.s.kuchenreuther@fau.de (I.K.); finn.clausen@fau.de (F.-N.C.); johanne.mazurie@fau.de (J.M.); franziska.czubayko@uk-erlangen.de (F.C.); anke.mittelstaedt@uk-erlangen.de (A.M.); ann-kathrin.koch@uk-erlangen.de (A.-K.K.); alara.karabiber@uk-erlangen.de (A.K.); frederik.hansen@med.uni-duesseldorf.de (F.J.H.); lisa-sophie.arnold@fau.de (L.-S.A.); nadine.weisel@uk-erlangen.de (N.W.); susanne.merkel@uk-erlangen.de (S.M.); maximilian.brunner@uk-erlangen.de (M.B.); christian.krautz@uk-erlangen.de (C.K.); robert.gruetzmann@uk-erlangen.de (R.G.); paul.david@uk-erlangen.de (P.D.); 2Faculty of Medicine, Friedrich-Alexander-Universität Erlangen-Nürnberg (FAU), 91054 Erlangen, Germany; 3Department of Dermatology, University Hospital Erlangen, 91054 Erlangen, Germany; sushmita.paul@uk-erlangen.de (S.P.); julio.vera-gonzalez@uk-erlangen.de (J.V.); 4Deutsches Zentrum für Immuntherapie, Friedrich-Alexander-Universität Erlangen-Nürnberg and University Hospital Erlangen, 91054 Erlangen, Germany; 5Bavarian Cancer Research Center (BZKF), 91052 Erlangen, Germany

**Keywords:** HVEM, inhibitory molecules, biomarkers, PDAC

## Abstract

Pancreatic ductal adenocarcinoma (PDAC) is aggressive, with a 5-year survival rate of only 12.8%, and its increasing incidence in Western countries highlights the urgent need for better early-stage detection and treatment methods. Early diagnosis significantly improves the chances of survival, but non-specific symptoms and undetectable precursor lesions pose a major challenge. To date, there are no reliable screening tools to detect PDAC at an early stage. Herpesvirus entry mediator (HVEM) has already been proposed as a prognostic marker in numerous cancer types. Therefore, we investigated the role of HVEM in PDAC. Flow cytometry was used to analyze HVEM expression in immune cells and its inhibitory receptors (CD160 and BTLA) on T-cells, as well as its subsets in the peripheral blood of 57 diagnosed PDAC patients and 17 clinical controls. In addition, survival analyses were performed within the PDAC cohort, changes in HVEM expression were analyzed in relation to clinicopathological parameters, and a correlation analysis between HVEM expression and cytokine levels of IL-6 and IL-10 was conducted. Furthermore, HVEM expression on monocytes and their subsets was evaluated as a potential prognostic marker and compared with the prognostic utility of CA19-9. We found that HVEM expression is significantly elevated on immune cells, particularly on monocytes (*p* < 0.0001) and their subsets, in PDAC patients, and is associated with reduced survival (*p* = 0.0067) and clinicopathological features such as perineural, lymphovascular, and vascular invasion. Moreover, HVEM-expressing monocytes demonstrated superior predictive value compared to CA19-9, highlighting their potential as part of a combined screening tool for PDAC. In conclusion, HVEM on monocytes could serve as a novel prognostic marker for PDAC.

## 1. Introduction

Pancreatic ductal adenocarcinoma (PDAC) is one of the most aggressive and deadly malignancies, with current therapeutic approaches proving largely inadequate. It is one of the fifteen most common cancers worldwide and has particularly high incidence and mortality rates in Western countries, especially in Europe and North America [1]. Projections based on WHO data from 2022 indicate a significant increase in the number of cases in the coming years, with an expected increase of 34.7% in Europe and 57.9% in North America by 2050. This alarming trend underlines the urgent need for early detection and more effective treatment options [2]. The prognosis for PDAC remains grim, with a 5-year overall survival rate of 12.8%. However, the 5-year relative survival depends heavily on the stage at which the disease is diagnosed. Patients with localized tumors confined to the pancreas have a 5-year survival rate of 44%. In comparison, patients with regional disease, which means that the tumor has invaded the lymph nodes, have a lower 5-year survival rate of 16.2%, while patients with distant metastases, usually of the liver or lungs, have the worst survival rate of only 3.1%. These statistics underline the crucial importance of early detection to improve the survival chances of patients with PDAC [3].

A major challenge in diagnosing PDAC is the absence of symptoms in its localized stages or the presence of only mild, nonspecific symptoms, even in metastasized stages. This makes symptom-based detection unreliable. Pancreatic cancers are categorized by the tissue they originate from, with PDAC being the most common type, arising from the exocrine pancreas [4]. The majority of PDACs develop from noninvasive microscopic precursor lesions known as pancreatic intraepithelial neoplasias (PanINs), which accumulate genetic alterations over time. These neoplasms are categorized into low-grade and high-grade based on histological criteria, as the morphological grade of dysplasia correlates with the risk of progression to invasive carcinoma [5,6]. Unfortunately, these precursor lesions cannot be detected through standard abdominal imaging techniques. In contrast, other precursor lesions, such as mucinous neoplasms, can be visualized using imaging methods. However, they account for only 10% of cases [7]. The treatment options for PDAC are currently still limited; treatment depends largely on the stage of the tumor. For localized tumors, surgical resection is the only curative option, followed by adjuvant chemotherapy with mFOLFIRINOX in fit patients or gemcitabine and capecitabine in less fit patients. For metastatic disease, treatment focuses on symptom control and palliative chemotherapy to improve the quality of life [8]. This highlights the critical need for the development of more effective early detection markers to identify PDAC at a stage when it is still treatable. CA19-9, one of the best-studied biomarkers for PDAC, is commonly used to diagnose symptomatic patients, assess prognosis and survival, and monitor the response to chemotherapy. However, its function as a screening tool is limited [9] and it would therefore be more useful when combined with other biomarkers [10].

In recent years, tumor immunology has become increasingly important. Checkpoint inhibitors that reverse T-cell suppression are now standard therapies for various solid tumors such as non-small cell lung cancer or malignant melanoma. Unfortunately, in PDAC, immunotherapy remains limited due to the immunosuppressive tumor microenvironment, the dense extracellular matrix, and a lack of reliable biomarkers [11]. Established immune checkpoints such as the PD-1/PD-L1 signaling pathway and the CTLA-4 axis have shown limited efficacy in PDAC, although several clinical trials have investigated their use either alone or in combination with other therapies [12]. Due to its coinhibitory role, herpesvirus entry mediator (HVEM) has attracted increasing attention in recent years as a potential target for cancer immunotherapy or as a prognostic marker [13]. HVEM, a member of the tumor necrosis factor superfamily, is mainly expressed in hematopoietic cells and interacts with five ligands: LIGHT, Lymphotoxin-alpha (LTa), B- and T-lymphocyte attenuator (BTLA), CD160, and SALM5 [14,15]. While HVEM has broader expression, BTLA is predominantly expressed in immune cells, especially within the lymphoid lineage. CD160 expression is even more restricted, being limited mainly to cytotoxic cells and T-cells [16]. The HVEM/BTLA pair forms a ligand–receptor network that is capable of bidirectional signaling. Under homeostatic conditions, HVEM and BTLA interact in the cis configuration on T-cells, initiating inhibitory signaling via the ITIM module in BTLA which recruits the tyrosine phosphatases SHP1/2 and limits antigen receptor activation [17]. When T-cells are activated, the expression of LIGHT increases, which leads to the internalization of HVEM. This allows BTLA to interact in trans with surrounding HVEM-expressing cells [18]. In trans, HVEM/BTLA signaling not only inhibits T-cells but also activates the HVEM-TRAF2 ligase pathway, promoting cell survival and differentiation in HVEM-expressing cells via NF-κB [19]. However, BTLA does not only act as an inhibitory receptor. In some cases, when stimulated by HVEM, it also delivers a pro-survival signal to the T-cell via a specific domain containing a GRB2 recognition motif [20]. Similarly, the HVEM/CD160 interaction triggers inhibitory signals in T-cells [16]. Although the interactions between HVEM and its ligands are very complex, they represent a potential target for immunotherapy or as a prognostic marker. Several studies have already shown that HVEM and its receptors play a role in autoimmune diseases [21,22] and in various solid tumors such as gastric cancer [23], hepatocellular carcinoma [24], melanoma [25], and many others, but little is known about HVEM and its ligands in PDAC. Therefore, in this study, we focused on HVEM and its receptors, BTLA and CD160, to investigate their roles in the context of PDAC.

## 2. Results

### 2.1. HVEM Expression on Circulating Leucocytes in Clinical Control and Gastrointestinal Cancer Patients

First, various immune cell types were characterized using flow cytometry. The gating strategy is detailed in Figure 1A. We identified neutrophils as CD15^+^HLA-DR^−^-cells, B-cells as CD19^+^HLA-DR^+^, monocytes as CD14^+^HLA-DR^+^, myeloid-derived suppressor cells (MDSCs) as CD14^+^HLA-DR^−^, and plasmacytoid dendritic cells (pDCs) as CD123^+^HLA-DR^+^. The mean fluorescence intensity (MFI) of HVEM in these cell populations was measured and patients with PDAC, esophageal carcinoma, and gastric carcinoma were compared with clinical controls. The PDAC patients showed a significantly increased expression of HVEM in MDSCs (*p* = 0.0004) (Figure 1B) as well as on neutrophils (*p* = 0.0002) (Figure 1C), pDCs (*p* < 0.0001) (Figure 1D), B cells (*p* = 0.0025) (Figure 1E), and monocytes (*p* < 0.0001) (Figure 1F) compared to the clinical controls, while this increase was not observed in patients with esophageal or gastric cancer. As the difference was most pronounced in monocytes, we focused on monocytes in particular in the following analysis. Next, we grouped the PDAC cohort into low and high HVEM expression groups based on the median (MFI = 201) and potential differences in patient survival were assessed. Patients with high HVEM expression on monocytes showed significantly shorter survival proportion in comparison to those with low expression of HVEM (*p* = 0.0067; HR = 6.103; 95% CI = [2.055; 18.13]) (Figure 1G). In addition, monocytes with elevated HVEM expression were associated with a 7.944-fold higher risk of death compared to those with low HVEM expression (Figure 1H), suggesting that HVEM-expressing monocytes have a negative impact on patient prognosis. Additionally, the PDAC cohort was further subdivided based on clinicopathological parameters to determine potential distinctions between early and late tumor stages. In patients with inoperable tumors, the advanced stage of the disease prevented surgical removal, and therefore, pathological staging was not conducted. For the purposes of analysis, these patients were classified as having advanced tumor stages. While the parameters pT, pN, cM/pM, G, and neoadjuvant chemotherapy showed no significant differences (Figure A1), it was observed that the MFI of HVEM on monocytes was significantly elevated in patients with lymphovascular (L1) (*p* = 0.0083), vascular(V1) (*p* = 0.0071), and perineural invasion (Pn1) (*p* = 0.0387) in comparison with patients without lymphovascular (L0), vascular (V0), and perineural invasion (Pn0) (Figure 1I). Since these parameters are associated with poorer survival and are independent risk factors [26,27,28], the increase in HVEM expression in these patients may contribute to the observed negative prognosis. This suggests that HVEM overexpression on monocytes could play a role in promoting more aggressive disease features, thereby contributing to the overall reduced clinical outcomes in these patients. To investigate a possible association between HVEM expression on monocytes and the proinflammatory cytokine interleukin-6 (IL-6) or the anti-inflammatory cytokine interleukin-10 (IL-10), we performed cytokine analysis of plasma samples from our PDAC and control cohorts. Our results showed that the levels of IL-6 and IL-10 were elevated in PDAC patients compared to the clinical controls. However, we did not find a significant correlation between the MFI of HVEM and the plasma concentrations of IL-6 or IL-10 in these patients (Figure A2). In summary, HVEM expression was significantly increased in several cell types compared to the controls, with the increase being most pronounced in monocytes. Higher HVEM expression in PDAC patients correlated with reduced survival and a higher risk of death. In addition, HVEM levels on monocytes were particularly high in PDAC patients with invasive features.

### 2.2. HVEM Expression on Monocyte Subsets Correlates with Worst Outcomes

Monocytes can be classified into three distinct subsets based on the expression of CD14 and CD16, each with unique functional roles. These subsets include CD14^++^CD16^−^ classical monocytes (CMs), CD14^++^CD16^+^ intermediate monocytes (IMMs) and CD14^+^CD16^++^ non-classical monocytes (NCMs) [29,30]. In this study, these monocyte subsets were identified and the MFI of HVEM was measured on each subset (Figure A3A,B). The results demonstrated a significantly elevated expression of HVEM on all monocyte subsets in patients with PDAC when compared to clinical controls. The *p*-values for HVEM expression were as follows: (*p* < 0.0001) for CMs, (*p* < 0.0001) for IMMs, and (*p* < 0.0001) for NCMs. Furthermore, survival analysis was conducted, as previously described, for each monocyte subset with the median MFI of the patients: (175) for CMs, (326) for IMMs, and (203) for NCMs. Notably, in all subsets, patients exhibiting high HVEM expression had significantly shorter survival times compared to those with low HVEM expression. While CMs and IMMs showed the same results (*p* = 0.0067; HR = 6.103; 95% CI = [2.055; 18.13]), the most significant difference in survival was observed in the NCM subgroup (*p* = 0.0007; HR = 14.01; 95% CI = [4.709; 41.69]). The odds ratio was calculated for all subsets to evaluate the risk of death associated with low versus high HVEM expression. Patients with high HVEM expression on CMs and IMMs exhibited a 7.944-fold elevated risk of death compared to low expression, whereas those with high HVEM expression on NCMs faced an even greater risk, with a 19.059-fold increase (Figure 2). Finally, the PDAC study cohort was again subdivided based on the clinicopathological parameters of the patients. While the parameters pT, pN, pM, G, Pn, and neoadjuvant chemotherapy showed no significant differences (Figure A1B–D), PDAC patients exhibiting lymphovascular and vascular invasion demonstrated significantly higher HVEM expression on monocyte subsets compared to those without such invasions. Particularly, in CMs, the difference was more pronounced in cases of vascular invasion, whereas in IMMs and NCMs, the difference was greater in cases of lymphovascular invasion (Figure 2). Characteristic features of the study population dichotomized by the median HVEM on monocytes, CMs, IMMs, and NCMs are listed in Table A1, Table A2, Table A3 and Table A4. These data suggest that monocytes and their subsets expressing HVEM are highly likely to play a major role in the progression of PDAC and influence the prognosis of patients. To investigate a possible association between HVEM expression on monocyte subsets and IL-6 or IL-10, we also correlated, as described above, the HVEM MFI of our PDAC study cohort with the corresponding plasma cytokine levels of IL-6 and IL-10. Yet, we did not find significant differences between the MFI of HVEM and the plasma concentrations of IL-6 or IL-10 in these patients (Figure A2). To ensure that our results were driven by the tumor itself and not by inflammatory events, we analyzed the correlations between the MFI of HVEM on monocytes and their subsets and inflammatory markers, including leucocyte count, C-reactive protein (CRP), and bilirubin. No significant correlation was found between the HVEM MFI and these inflammatory markers (Figure A6). In conclusion, HVEM expression was significantly elevated in CMs, IMMs, and NCMs, with higher levels in PDAC patients linked to poorer survival and increased risk of death, especially for NCMs. Additionally, PDAC patients with lymphovascular and vascular invasion showed particularly high HVEM levels on monocytes.

### 2.3. Alterations in the Frequency of BTLA- and CD160-Expressing T-Cells in PDAC

Next, our focus was on identifying changes in the frequencies of the HVEM receptors (BTLA and CD160) on T-cells. The gating strategy for this can be seen in Figure 3A: We first gated for T-cells (CD3^+^), then classified the T-cells into cytotoxic T-cells (CD8^+^), T-helper-cells (CD4^+^), and double-negative T-cells (CD4^−^CD8^−^). Next, we characterized effector T-cells as CCR7^−^CD45RA^+^ (TE), naive T-cells as CCR7^+^CD45RA^+^ (NT), central memory T-cells as CCR7^+^CD45RA^−^ (TCM), and effector memory T-cells as CCR7^−^CD45RA^−^ (TEM) within the helper and cytotoxic T-cell populations. First, we compared the frequencies of TEM, TE, NT, and TCM cells in CD4^+^ and CD8^+^ cells between PDAC patients and clinical controls. We found a significantly increased frequency of CD4^+^ TCM cells (Figure A4A). No significant changes were observed in other T-cell subtypes (Figure A4). In all mentioned cell populations, we also determined the frequencies of BTLA^+^ and CD160^+^ cells, comparing PDAC patients with clinical controls. This comparison is shown as exemplary in the gating strategy for CD4^+^ TEM cells (Figure 3A). A significantly higher frequency of CD160^+^ cells was found in CD8^+^ NT (Figure 3B) and CD8^+^ TCM cells (Figure 3C) in PDAC patients. Additionally, BTLA^+^ cells were significantly decreased in CD8^+^ TCM cells (Figure 3C). No significant changes in BTLA and CD160 frequencies were observed in other cell populations (Figure A4 and Figure A5). As mentioned, the ligand–receptor network of HVEM-BTLA and HVEM-CD160 is very complex. Further research is needed to determine the exact roles of BTLA and CD160 in the context of PDAC.

### 2.4. HVEM Expression on Monocyte Subsets Correlates with the Phagocytic Activity and Cytokine Production During Pancreatic Cancer

In order to comprehend the functional role of monocytes in pancreatic cancer, we performed a coculture of monocytes with the human pancreatic cancer cell line Panc-1 (Figure 4A). Initially, we analyzed the expression of HVEM on different monocyte subsets and found that all three subsets (CMs, IMMs, and NCMs) were significantly increased in monocytes cultured with Panc-1 compared to those cultured without Panc-1 (Figure 4B). Monocytes are known to be phagocytic, so we subsequently examined the phagocytic property of the monocyte subsets. Our findings revealed a significant decrease in the phagocytic activity of classical monocytes when cultured with Panc-1 compared to those cultured without Panc-1. However, no difference in the phagocytic activity of intermediate and non-classical monocytes was observed in IMMs and NCMs cultured with Panc-1 compared to those cultured without Panc-1 (Figure 4C). Furthermore, the expression of HVEM on the subsets influenced the phagocytic property of monocytes. Monocytes also have an important function in producing cytokines. However, no differences in the levels of IL-6 and IL-10 were observed (Figure 4E). However, a significant increase in TNF-a was observed in monocytes cultured with Panc-1 cells compared to those cultured without Panc-1. Taken together, our data show the detrimental role of HVEM-expressing monocytes during pancreatic cancer.

### 2.5. Discriminating Power of HVEM-Expressing Monocytes and Their Subsets with Serum CA19-9 to Discriminate PDAC Patients from Clinical Control Patients

Finally, we investigated the potential of HVEM expression on monocytes as a prognostic marker and evaluated its predictive efficacy compared to the commonly used CA19-9 marker. Receiver operating characteristic (ROC) curve analyses were performed for HVEM-expressing monocytes plus subsets and CA19-9 levels with a comparison of their area under the curve (AUC) values. These analyses revealed that the AUC values for HVEM-expressing monocytes and subsets, especially for NCMs, were higher than those for CA19-9, indicating a higher predictive value for HVEM-expressing NCM (Figure 5A). Furthermore, generalized linear models using the ROCit package were applied to evaluate combinations of HVEM-expressing monocytes, CMs, IMMs, and NCMs with CA19-9 values. This combined approach resulted in an improved classification model compared to single markers, with the combination of HVEM-expressing NCMs and CA19-9 achieving the highest AUC value of 0.872 (Figure 5B). We previously found that patients with high HVEM expression have shorter survival than patients with low expression, with this observation being most pronounced in the NCM subgroup. Based on these results, we identified HVEM-expressing NCM as the most promising prognostic marker. To investigate whether a combination of more than two markers provides better results, we created two additional ROC curves: one containing all monocytes and another combining all markers, including CA19-9. The combination of all markers achieved the highest AUC value of 0.882 (Figure 5C), a result consistent with the expectation that the inclusion of comprehensive patient information improves predictive ability. For an overview of the AUC values, optimal cut-off points, and corresponding sensitivity and specificity values, see Figure 5D. To assess whether HVEM, in combination with other biomarkers, has greater discriminative power than the biomarkers alone, we combined HVEM on monocytes (subsets) with the patients’ bilirubin, CEA, and CRP levels. In all cases, this resulted in improved predictive power, supporting our hypothesis that HVEM, in combination with other biomarkers, has the potential to serve as a diagnostic tool in the future (Figure A7). In summary, HVEM-expressing monocytes have high potential as prognostic markers, with their predictive efficacy further enhanced in combination with other biomarkers.

## 3. Discussion

We demonstrated that, in PDAC, HVEM expression in immune cells is significantly elevated, particularly on monocytes and their subsets, CMs, IMMs, and NCMs. Furthermore, patients with high HVEM expression were associated with poorer survival outcomes and an increased risk of death. HVEM expression on monocytes was also associated with clinicopathological features, including perineural invasion, lymphovascular invasion, and vascular invasion. We not only found changes in HVEM expression, but also in the frequency of T-cells expressing the receptors of HVEM. Compared to CA19-9, HVEM-expressing monocytes showed superior predictive value, suggesting that a combined analysis of these two markers could serve as a potential screening tool for PDAC in the future. We hypothesize that HVEM-expressing NCMs play a crucial role, as patients with high HVEM expression on NCMs exhibited the worst outcomes, and this marker proved to be the most reliable prognostic marker in our comparison.

Monocytes, part of the mononuclear phagocyte system, develop from hematologic precursors in the bone marrow. After entering the bloodstream, they fulfill several functions in the body, for example, in antigen presentation as part of the adaptive immune system, or as precursors for other immune cells such as dendritic cells, MDSCs, or macrophages, to name but a few [31]. In the case of PDAC, both monocyte-derived macrophages and embryonically derived macrophages belong to tumor-associated macrophages (TAMs) which are associated with disease progression, metastasis, and shorter survival [32]. A study revealed that an inflammatory interplay between IL-1β-expressing TAMs and pancreatic cancer cells drives disease progression and that targeting IL-1β activity could reprogram the immune microenvironment and control tumor growth [33]. Another study showed that cancer-associated fibroblasts convert TAMs to an immunosuppressive M2-like phenotype and limit FOLFIRINOX-induced cell death [34].

In pancreatic cancer, we were able to show that PDAC patients had significantly increased levels of monocytes in general in the peripheral blood. This increase was associated with more aggressive tumor growth, poorer survival outcomes, and the elevated activation status of the respective monocytes [30]. Moreover, another study showed that so-called inflammatory monocytes (IMs) are increased in the blood of pancreatic cancer patients but reduced in the bone marrow, with a higher ratio of IMs in the blood to IMs in the bone marrow serving as a new predictor of poorer survival after tumor resection. Human pancreatic cancer tumors produce CCL2, which attracts immunosuppressive CCR2+ monocytes that infiltrate the tumor microenvironment and differentiate into TAMs. Patients with tumors that have high CCL2 levels and low CD8+ T-cell infiltration have significantly lower survival. In mouse models, the inhibition of CCR2 leads to a reduction in IMs and macrophages in primary tumors and in the pre-metastatic liver, thereby enhancing anti-tumor immunity, reducing tumor growth, and limiting metastasis [35]. The lymphocyte–monocyte ratio (LMR) has recently attracted increasing interest as it represents the immunological situation during cancer, with lymphopenia as a marker of poor immune response and an increased monocyte count as a marker of high tumor burden [36]. In pancreatic cancer, a high LMR before treatment is linked to longer disease-free survival and recurrence-free survival, and a shorter time to disease progression [37]. These findings support our hypothesis that monocytes play an important role in the progression of PDAC.

Monocytes can be further subdivided into CMs (CD14^++^CD16^−^), IMMs (CD14^++^CD16^+^), and NCMs (CD14^+^CD16^++^) [38]. One study found that IMMs and NCMs were elevated in human blood during sepsis, an acute inflammatory response, while only NCMs increased in patients with systemic lupus erythematosus, a chronic inflammatory condition, suggesting that IMM levels could help differentiate between acute and chronic inflammation. Additionally, the roles of monocyte subsets were characterized: CMs are primarily phagocytic, NCMs are involved in inflammation and antigen presentation, and IMMs perform both phagocytic and inflammatory functions [39]. Together with our findings, we assume that NCMs play a key role not only in acute and chronic inflammation, but also in cancer. HVEM and its inhibitory receptors BTLA and CD160 have become the focus of intensive tumor research as a potential new therapeutic target or prognostic marker [18]. As an immune checkpoint molecule, HVEM provides bidirectional signals during T-cell activation. While the HVEM/LIGHT pathway enhances the eradication of tumors, the binding of HVEM to BTLA, which is predominantly expressed in tumor-specific T-cells, suppresses the response of CD8+ T-cells and thus promotes immune evasion. It was found that high HVEM expression is associated with a low number of tumor-infiltrating T-cells and reduced expression of interferon-γ, perforin, and granzyme B. Knocking out the HVEM gene improves the immune response, inhibits the proliferation of tumor cells and increases the sensitivity of T-cells in certain types of cancer. In addition, HVEM can directly promote tumor growth, as its silencing reduces proliferation in esophageal and renal cancer [40]. In colorectal cancer, BTLA was significantly upregulated, which was associated with advanced tumor stage and a shorter survival time. Due to its predictive power to differentiate colorectal cancer patients from control patients, it has been proposed as a prognostic marker [41]. Lan, X., et al. found a correlation between HVEM and BTLA expression during gastrointestinal cancer. In addition, the HVEM and BTLA expression was linked to poorer overall survival and characteristics of tumor progression such as the depth of invasion, lymph node metastasis, and the histological grade [23]. In another study, a correlation was found between HVEM expression, the depth of tumor invasion, and lymph node metastasis in esophageal cancer. In addition, HVEM expression was found to result in fewer tumor-infiltrating CD4^+^, CD8^+^, and CD45RO^+^ lymphocytes. HVEM gene silencing led to the reduced proliferation of cancer cells. This antitumor effect was associated with an upregulation of the local immune response, suggesting that HVEM could present a new therapeutic target [42]. In breast cancer, patients with low tumor-infiltrating lymphocytes in HVEM-positive tumors seemed to have the worst outcome. Moreover, HVEM expression was most prevalently found in tumors of the human epidermal growth factor receptor 2-overexpressed subtype [43]. A study on HVEM expression in malignant melanoma came to similar conclusions and hypothesized that HVEM targeting could be used, especially in a metastatic state [25]. Several other cancer studies have reported alterations in HVEM expression, often linked to poor prognosis in patients with various malignancies, such as hepatocellular carcinoma [44], chronic lymphocytic leukemia [45], clear cell renal cell carcinoma [46], and glioblastoma [47]. These studies support our findings and collectively suggest that HVEM and its receptors play a significant role in tumor progression across different cancer types.

In a mouse model of FV infection, David P. et al. showed that HVEM expression increased in infected myeloid cells, while CD8+ T-cells showed elevated BTLA and CD160 levels after infection. Additionally, a combined blockade of PD-L1, CD160, and HVEM enhanced T-cell function and reduced viral load [48]. Another study demonstrated that the competitive blocking of HVEM to BTLA but not to LIGHT suppresses immune rejection in a murine GvHR model, indicating that effective CD4 and CD8 T-cell responses rely on HVEM/BTLA signaling and that the HVEM/BTLA pathway could present a new therapeutic target in autoimmune diseases [49]. Moreover, Guruprasad P. et al. found that the BTLA/HVEM signaling axis suppresses CAR T-cell activity within the tumor microenvironment in several models of HVEM+ malignancies. Deleting BTLA from CAR-T-cells enhanced their anti-tumor function and allowed these cells to withstand immunosuppressive influences from regulatory T-cells and macrophages. The study suggested BTLA suppression as a promising target in CAR T-cell therapies [50]. The BTLA/HVEM axis has already been targeted in vitro and in vivo to study its effects on tumors. A study by Chen Y. et al. found that the combination of chemotherapy with an anti-BTLA antibody that inhibits IL-6/IL-10-induced CD19-high B-lymphocytes significantly reduced peritoneal tumor volume and prolonged the survival time of mice with epithelial ovarian carcinoma [51]. In another study, a synergistic effect between the PD-1 blockade and other immune markers such as TIM-3, BTLA, LAG-3, and CTLA-4 was discovered. This combination promoted the proliferation of T-cells and the production of cytokines, which is due to the upregulation of co-inhibitory T-cell receptors after a PD-1 blockade [52].

**Limitations:** Against this background, we formulate the limitations of our study. Our main focus was on HVEM as the marker of interest. Further research is needed to investigate the interaction between HVEM and its receptors BTLA and CD160 in PDAC and to determine whether blocking the interactions results in antitumor immunity as seen in the studies named above. In addition, we were only able to enroll a small number of patients, which affected the power of our study. As all patients in our study were over 40 years old, many of them had multiple comorbidities, which also meant thar the majority of patients took numerous medications. Although care was taken to ensure that there were no concurrent tumor diagnoses, other pre-existing conditions and medications may still have influenced the results. Additionally, some patients may have had underlying diseases which had not yet been diagnosed. Furthermore, our study was conducted in a specialized pancreatic cancer center. Therefore, our patient population may include more complex cases than if the same study had been conducted in a smaller hospital. Since the control patients also had a need for surgery, they did not represent healthy controls but rather clinical controls. In this respect, the PDAC and control groups are similar in that surgery was required in both cases. However, a possible influence of the current diagnosis of the control patients on the study results cannot be excluded. Finally, we focused primarily on monocytes. Since HVEM is also expressed by other cells, it is likely that these could also play a role and be involved in the tumor environment, and could also serve as prognostic markers. It is therefore important to conduct further studies to determine whether HVEM can also be used in a clinical setting in the future.

## 4. Materials and Methods

### 4.1. PDAC and Control Cohort

Preoperative peripheral blood samples were obtained from patients of both genders aged 40 years or older who underwent elective surgery with a suspicion of PDAC. Patients with a postoperative confirmation of PDAC were included in the study. As clinical controls, we included patients of the same age criterion and blood collection time who did not have a recent cancer diagnosis. The indications for surgery in the control group included hernia, cholecystectomy, or obesity. The sample collection occurred between 2020 and 2024 at the Department of Surgery, University Hospital Erlangen, Germany, resulting in a total of 57 PDAC patients and 17 controls. The study was conducted in accordance with the Declaration of Helsinki and was approved by the Institutional Review Board of the University Hospital Erlangen (Nr. 180_19 B, 14.06.2019). The study patients or their legal guardians signed the written informed consent form prior to surgery. The following tables illustrate the characteristics of the PDAC and Control cohorts (Table 1 and Table 2).

### 4.2. Sample Processing

Blood samples were collected in 7.5 mL EDTA tubes (Cat-No.01.1605.001, Sarstedt, Nümbrecht, Germany) and processed on the same day. The blood was first centrifuged at 350× *g* for 10 min at room temperature. The plasma was then pipetted off and stored at −80 °C for subsequent plasma cytokine analysis. The remaining cells were treated with a 1:10 dilution of 1× RBC lysis buffer (Cat-No. 555899, BD Biosciences, Franklin Lakes, NJ, USA) and incubated for 15 min at room temperature. Following a second centrifugation at 350× *g* for 5 min the cell pellet was resuspended in 50 mL of 1× PBS (Cat-No. 14190169, Gibco, Waltham, MA, USA). After another centrifugation at 350× *g* for 5 min, the cells were resuspended in FACS buffer (PBS containing 1% FBS (Cat-No. A3160802, Gibco, Waltham, MA, USA), 0.5% BSA (Cat-No. A2153, Sigma Aldrich, St. Louis, MO, USA), and 2 mM of EDTA (Cat-No. AM9260G, Invitrogen, Waltham, MA, USA)) for surface FACS staining. The cell suspension was transferred to 96 well plates (Cat-No. 651101, Greiner BIO-ONE, Kremsmünster, Austria) and washed with FACS buffer. Afterward, antibodies were added, and the cells were incubated at four degrees for 20 min. After incubation, the cells were washed twice with FACS buffer. Finally, the cells were resuspended in FACS buffer and acquired on a Celesta (BD Biosciences, Franklin Lakes, NJ, USA) flow cytometer using the BD FACSDiVa™ software v8.0.1.1 and analyzed with FlowJo (Version 10.9.0 for Windows, FlowJo Software, Ashland, OR, USA).

### 4.3. Flow Cytometry

Three staining panels were applied for flow cytometric analysis. One to determine HVEM levels on different immune cells, one to analyze the frequency of CD160 and BTLA expressing T-cells, and one for the functional analysis of monocytes. The following antibodies were used: Anti-HLA DR-BUV395 (Cat-No. 564040, BD Biosciences, Franklin Lakes, NJ, USA), Anti-CD14-BUV737 (Cat-No. 612763, BD Biosciences, Franklin Lakes, NJ, USA), Anti-CD16-BV510 (Cat-No. 563830, BD Biosciences, Franklin Lakes, NJ, USA), Anti-CD270-BV786 (Cat-No. 743827, BD Biosciences, Franklin Lakes, NJ, USA), Anti-CD19-FITC (Cat-No. 555412, BD Biosciences, Franklin Lakes, NJ, USA), Anti-CD3-PE (Cat-No. 555340, BD Biosciences, Franklin Lakes, NJ, USA), Anti-CD19-PE (Cat-No. 555413, BD Biosciences, Franklin Lakes, NJ, USA), Anti-CD20-PE (Cat-No. 555623, BD Biosciences, Franklin Lakes, NJ, USA), Anti-CD56-PE (Cat-No. 555516, BD Biosciences, Franklin Lakes, NJ, USA), Anti-CD123-PE-Dazzle (Cat-No. 562391, BD Biosciences, Franklin Lakes, NJ, USA), Anti-CD15-BV421 (Cat-No. 323040, Biolegend, San Diego, CA, USA), Anti-CD11b-PeCy5 (Cat-No. 555389, BD Biosciences, Franklin Lakes, NJ, USA), Anti-CD11c-BV711 (Cat-No. 563130, BD Biosciences, Franklin Lakes, NJ, USA), and Anti-CD33-BV650 (Cat-No. 740573, BD Biosciences, Franklin Lakes, NJ, USA) for the HVEM analysis; Anti-CD107a-BUV395 (Cat-No. 565113, BD Biosciences, Franklin Lakes, NJ, USA), Anti-CD45RA-BUV737 (Cat-No. 612846, BD Biosciences, Franklin Lakes, NJ, USA), Anti-CCR7-BV510 (Cat-No. 563449, BD Biosciences, Franklin Lakes, NJ, USA), Anti-CD45-BV786 (Cat-No. 563716, BD Biosciences, Franklin Lakes, NJ, USA), Anti-CD160-Alexa Fluor488 (Cat-No. 562351, BD Biosciences, Franklin Lakes, NJ, USA), Anti-TCRgd-PE (Cat-No. 331210, Biolegend, San Diego, CA, USA), Anti-CD73-PE-Dazzle (Cat-No. 344020, Biolegend, San Diego, CA, USA), Anti-CD272-BV421 (Cat-No. 564802, BD Biosciences, Franklin Lakes, NJ, USA), Anti-CD4-PeCy5 (Cat-No. 560650, BD Biosciences, Franklin Lakes, NJ, USA), Anti-CD8-BV711 (Cat-No. 563676, BD Biosciences, Franklin Lakes, NJ, USA), and Anti-CD3-BV650 (Cat-No. 563852, BD Biosciences, Franklin Lakes, NJ, USA) for the T-cell analysis; and Anti-CD80-BUV395 (Cat-No. 565210, BD Biosciences, Franklin Lakes, NJ, USA), Anti-CD14-BUV737 (Cat- No. 612763, BD Biosciences, Franklin Lakes, NJ, USA), Anti-CD16-BV510 (Cat-No. 563830, BD Biosciences, Franklin Lakes, NJ, USA), Anti-CD45-BV786 (Cat-No. 563716, BD Biosciences, Franklin Lakes, NJ, USA), Anti-CD270-PE (Cat-No. 318805, Biolegend, San Diego, CA, USA), Anti-CD206-PE-Dazzle (Cat-No. 321129, Biolegend, San Diego, CA, USA), Anti-CD163-PeCy5 (Cat-No. 326511, Biolegend, San Diego, CA, USA), Anti-CCR2-BV711 (Cat-No. 357231, Biolegend, San Diego, CA, USA) and Anti-PDL1-BV650 (Cat-No. 563740, BD Biosciences, Franklin Lakes, NJ, USA) for the functional analysis of monocytes.

### 4.4. Enzyme-Linked Immunosorbent Assay (ESSAY)

Cytokine levels were determined by performing ELISA on the patients’ plasma samples with the ELISA MAX™ Deluxe Set for Human IL-6 (Cat-No. 430504, Biolegend, San Diego, CA, USA) and IL-10 (Cat-No. 430604, Biolegend, San Diego, CA, USA) from BioLegend. The ELISA procedure was carried out as follows. Initially, 100 µL of the capture antibody in coating buffer was added to each well of a 96-well plate. The plate was then incubated overnight at 4 °C. Subsequent to this step, the plate was washed and blocked. Thereafter, 100 µL of the suitably diluted standards and samples were added to the appropriate wells. The plate was then incubated at room temperature for 2 h, after which it was thoroughly washed. In subsequent steps, 100 µL of the diluted biotinylated detection antibody in assay diluent was added to each well, and the plates were incubated at room temperature for 1 h, after which they were washed. Then, 100 µL of diluted avidin–horseradish peroxidase in assay diluent was added to each well. This was incubated at room temperature for 30 min and was washed. Thereafter, 100 µL of TMB substrate was added to each well. This was left in the dark for 15 min. Finally, 100 µL of the stop solution was added, and an absorption reading at 450 nanometers was taken. Data acquisition was performed using an ELISA Plate Reader SpectraMax M3.

### 4.5. Human PBMC Isolation and Monocytes Functional Assay

All specimens were obtained from donor volunteers using 7.5 mL EDTA tubes (Cat-No.01.1605.001, Sarstedt, Nümbrecht, Germany). After collection, the tubes were promptly mixed gently up and down. PBMCs were isolated using the Ficoll density gradient (GE Healthcare, Little Chalfont, UK). Monocytes from human PBMCs were isolated using the plastic adhesion method [53]. Monocytes were stimulated for 4 days with 50 ng/mL of macrophage-colony-stimulating factor (M-CSF) in RPMI 1640 media. The cells were seeded in 24-well tissue culture plates (Thermo Fisher Scientific, Waltham, MA, USA) at a density of 1 × 10^6^ cells per ml. The monocytes were allowed to adhere and were differentiated into MDMs for 4 days under a 5.0% CO_2_ atmosphere at 37 °C. After four days, the supernatant was collected from each well and stored for subsequent cytokine analysis. To the wells, 1 mL of cold 1× PBS was added and the adherent cells were scraped off. The collected cell suspension was transferred into a 15 mL Falcon tube and centrifuged at 400× *g* for 4 min. The supernatant was discarded, and the pellet was resuspended in FACS buffer (PBS containing 1% FBS (Cat-No. A3160802, Gibco, Waltham, MA, USA), 0.5% BSA (Cat-No. A2153, Sigma Aldrich, St. Louis, MO, USA), and 2 mM EDTA (Cat-No. AM9260G, Invitrogen, Waltham, MA, USA)) for subsequent FACS analysis.

The supernatants from the cocultures were analyzed using LEGENDPlexTM bead-based immunoassays (Cat-No. 740527, Biolegend, San Diego, CA, USA) according to the manufacturer’s instructions. TNF-α, IL-6, and IL-10 were simultaneously quantified. Data acquisition was performed on flow cytometer and analyzed with the LEGENDPlex^TM^ Data Analysis Software Suite (https://legendplex.qognit.com/workflow/171050 (accessed on 17 March 2025), Biolegend, San Diego, CA, USA).

### 4.6. Statistical Analysis

Data were stored on a laboratory computer and analyzed using GraphPad Prism (Version 9.5.1 (733) for Windows, GraphPad Software, Boston, MA, USA). For comparisons between two independent groups, an independent t-test with Welch’s correction was applied and for comparing two related groups, a paired t-test was used. Survival data were evaluated using the log-rank test, and both the hazard ratio (HR) and the 95% confidence interval (95% CI) were reported. Correlation analysis was performed via simple linear regression, with the corresponding regression line plotted. Both two-tailed *p*-values and R^2^ values were reported. To examine whether the two categorical variables were independent, a chi-squared test was conducted. The statistical significance was set at *p* < 0.05. The odds ratio was determined using SPSS software (28.0.0.0 (190)) of IBM (SPSS Software, Armonk, NY, USA).

ROC plots, as presented in Figure 4, were generated using the ROCit package of the programming language R. In order to combine different markers, generalized linear models (GLMs) were fitted to the acquired data. Since the differentiation between PDAC and clinical control patients represents a two-class problem, a binomial distribution was selected for the optimization of the GLMs. In the following equations, $x_{1}$ to $x_{N}$ refer to the marker values of a specific patient (i.e., the MFI of HVEM on monocytes, CMs, IMMs, NCMs, and CA19-9 levels), while $\beta_{0}$ to $\beta_{N}$ are weighting coefficients obtained by the respective GLM. Against this background, Equation (1) describes the calculation of the prediction value $\hat{p}$, where σ denotes the sigmoid function which maps arbitrary input values to the range [0, 1]. Finally, the criterion presented in Equation (2) can be used to decide between PDAC or no PDAC. To determine the optimal cutoff value, the Youden index was calculated for each point in the ROC curve. The cutoff value with the highest Youden index was selected as the optimal one, with the corresponding sensitivity and specificity reported in Figure 4B.(1)$$\hat{p}=\sigma (\beta_{0}+\beta_{1}\;*\;x_{1}+\beta_{2}\;*\;x_{2}+\ldots +\beta_{N}\;*\;x_{N})$$(2)PDAC Prediction=   yes   no    if  p^≥cutoff if p^<cutoff

## 5. Conclusions

HVEM on monocytes is associated with a more aggressive tumor stage and a poorer prognosis. In the future, HVEM could be used as a stand-alone prognostic marker or combined with others to improve accuracy. Combining HVEM with established markers appears to be more promising, using patient blood samples and a generalized linear model to create a combined value for prognosis based on a cut-off value determined from a larger study. These findings also underscore the potential significance of the HVEM–monocyte axis as a therapeutic target during PDAC. Moreover, a transfer of analysis to other gastrointestinal cancer types could help to improve diagnostics for more patients.

## Figures and Tables

**Figure 1 ijms-26-02875-f001:**
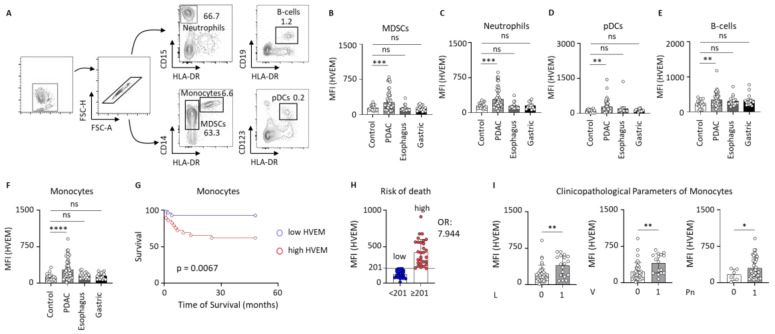
During PDAC, patients show increased HVEM expression in immune cells. Especially on monocytes, high HVEM expression is associated with a poorer prognosis and an increased risk of death, as well as with clinicopathological features. (**A**) A gating strategy to characterize B-cells (CD19^+^HLA-DR^+^), MDSCs (CD14^+^HLA-DR^−^), monocytes (CD14^+^HLA-DR^+^), neutrophils (CD15^+^HLA-DR^−^), and pDCs (CD123^+^HLA-DR^+^); a comparison of the MFI of HVEM on (**B**) MDSCs, (**C**) neutrophils, (**D**) pDCs, (**E**) B-cells, and (**F**) monocytes of clinical control patients (n = 17) and patients with PDAC (n = 57), esophageal carcinoma (n = 17), and gastric carcinoma (n = 15); (**G**) survival curve analysis of patients expressing low versus high levels of HVEM; (**H**) the MFI of HVEM (median) defining the risk of death (low vs. high); (**I**) a comparison of the HVEM MFI according to the pTNM stage in PDAC patients; * *p* < 0.05; ** *p* < 0.01; *** *p* < 0.001; **** *p* < 0.0001; HVEM: herpesvirus entry mediator; MDSCs: myeloid-derived suppressor cells; MFI: mean fluorescense intensity; PDAC: pancreatic ductal adenocarcinoma; pDCs: plasmacytoid dendritic cells.

**Figure 2 ijms-26-02875-f002:**
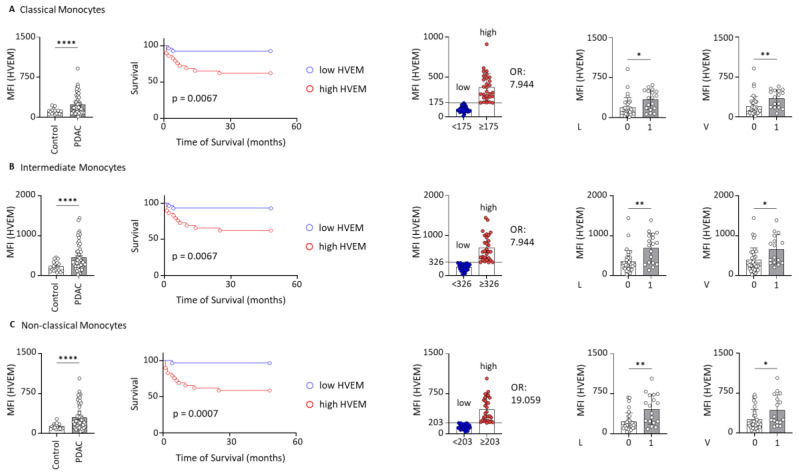
In PDAC patients, HVEM expression on CMs, IMMs, and NCMs is significantly increased compared to clinical controls. This upregulation of HVEM expression on monocyte subsets is associated with poorer overall survival and an increased risk of death, especially for NCMs. Increased HVEM expression is also linked to invasive clinical features, such as L and V. The figure illustrates the MFI of HVEM in PDAC and control patients, the survival curves of patients with low and high HVEM expression, the MFI of HVEM (median) defining the risk of death (low vs. high), and the MFI of HVEM in patients with and without lymphovascular invasion and vascular invasion for (**A**) classical monocytes, (**B**) intermediate monocytes, and (**C**) non-classical monocytes; * *p* < 0.05; ** *p* < 0.01; **** *p* < 0.0001; HVEM: herpesvirus entry mediator; L: lymphovascular invasion; MFI: mean fluorescence intensity; PDAC: pancreatic ductal adenocarcinoma; V: vascular invasion.

**Figure 3 ijms-26-02875-f003:**
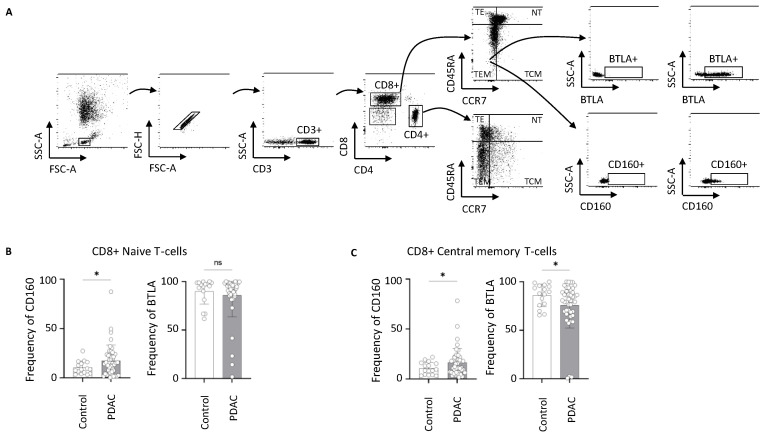
PDAC leads to changes in frequency of BTLA- and CD160-expressing NT and TCM cells. (**A**) Gating strategy to characterize effector T-cells (CCR7^−^CD45RA^+^), naive T-cells (CCR7^+^CD45RA^+^), central memory T-cells (CCR7^+^CD45RA^−^), and effector memory T-cells (CCR7^−^CD45RA^−^) among CD4+ and CD8+ T-cells; (**B**) frequency of CD160+ and BTLA+ cells among CD8+ naive T-cells of clinical control and PDAC patients; (**C**) frequency of CD160+ and BTLA+ cells among CD8+ central memory T-cells of clinical control and PDAC patients; * *p* < 0.05; BTLA: B- and T-lymphocyte attenuator; PDAC: pancreatic ductal adenocarcinoma; NT: naive T-cells; TE: effector T-cells; TCM: central memory T-cells; TEM: effector memory T-cells.

**Figure 4 ijms-26-02875-f004:**
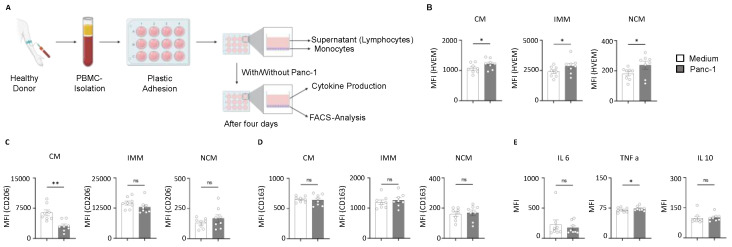
Functional role of monocytes during pancreatic cancer. (**A**) Schematic diagram denoting coculture system (Created in BioRender. Kuchenreuther, I. (2025) https://BioRender.com/e51y219 accessed on 19 February 2025). (**B**) Expression of HVEM on different monocyte subsets. (**C**) Expression of CD206 on different monocyte subsets. (**D**) Expression of CD163 on different monocyte subsets. (**E**) Cytokine levels of IL-6, TNF-a and IL-10 in monocytes with and without Panc-1. ** p* < 0.05; *** p* < 0.006; PBMC—peripheral blood mononuclear cells; CM—classical monocytes; IMM—intermediate monocytes; NCM—non-classical monocytes; IL-6—interleukin 6; TNFa—tumor necrosis factor alpha; IL10—interleukin 10.

**Figure 5 ijms-26-02875-f005:**
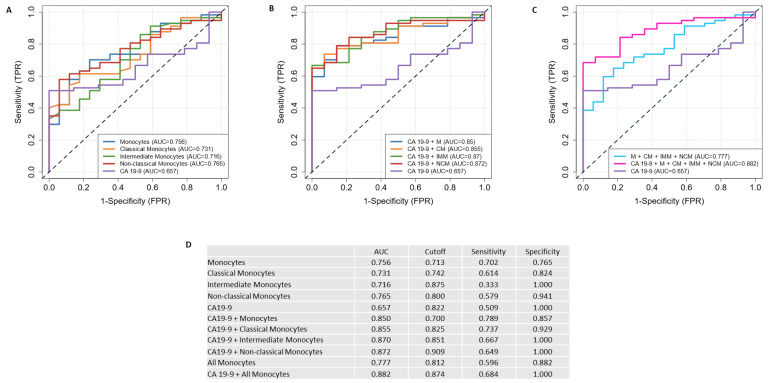
HVEM on monocytes (subsets) exhibits greater discriminative power than CA19-9, which can be enhanced through the combination of both markers. The combination of HVEM with other biomarkers could serve as a novel prognostic tool in the future. (**A**) ROC plots of single markers; (**B**) ROC plots of monocyte (subsets) combined with CA19-9; (**C**) ROC plots of all markers combined; (**D**) an overview of the AUC, optimal cutoff, sensitivity, and specificity of the ROC curves. AUC: area under the curve; HVEM: herpesvirus entry mediator; ROC: receiver operating characteristic.

**Table 1 ijms-26-02875-t001:** The clinico-pathological parameters of the PDAC study cohort.

		PDAC Patients
Number		57
Mean age (in years [range])		70 (43–90)
Sex (male:female)		24:33
pT category	(y)pT1	3
	(y)pT2	18
	(y)pT3	21
	(y)pT4	4
	Unresectable	11
pN category	(y)pN0	17
	(y)pN1,2	29
	Unresectable	11
Vascular invasion	V0	36
	V1	10
	Unresectable	11
Lymphatic invasion	L0	34
	L1	12
	Unresectable	11
Perineural invasion	Pn0	6
	Pn1	40
	Unresectable	11
Grading	G2	12
	G3	26
	Unresectable/neoadjuvant treatment	19
Distant metastasis	No	50
	Yes	7
UICC stage	(y)I	9
	(y)II	17
	(y)III	18
	(y)IV	7
	Unresectable	6
Neoadjuvant treatment	No	39
	Yes	18
ASA score	1	1
	2	13
	3	41
	Unresectable	2

**Table 2 ijms-26-02875-t002:** The clinico-pathological parameters of the control cohort.

	Clinical Control Patients
Number	17
Mean age (in years [range])	68 (39–85)
Sex (male:female)	7:10
Adipositas	1
Cholecystectomy	3
Hernia	7
Lipoma	1
Diaphragmatic hernia	5

## Data Availability

Data is contained within the article. The data presented in this study are available on request from the corresponding author.

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
