# Peer review of "Increased Herpesvirus Entry Mediator Expression on Circulating Monocytes and Subsets Predicts Poor Outcomes in Pancreatic Ductal Adenocarcinoma Patients"

_ijms, 2025, doi:10.3390/ijms26072875_

Round 1
Reviewer 1 Report
Comments and Suggestions for Authors
The manuscript analyzes the prognostic role of HVEM expression on monocytes and their various subsets in pancreatic ductal adenocarcinoma, PDAC. Authors demonstrate its correlation with reduced survival, aggressive clinicopathological features, and significant predictive value compared to CA19-9 as a biomarker. The study employs flow cytometry and survival analysis, highlighting the potential of HVEM as a novel biomarker for early detection and improved prognosis in PDAC.
In terms of originality and novelty, the research provides a novel perspective by focusing on HVEM expression in PDAC, which is less explored compared to other cancers. In addition its innovative approach to comparing HVEM to CA19-9 enhances the overall significance. Scientifically, the study is sound and easy to follow, with robust methodologies and detailed statistical analyses. Overall, this manuscript contributes to the field of cancer biomarker research and offers (potential) translational applications for PDAC prognosis and screening. Some minor comments below:
1. The study demonstrates association between HVEM expression and survival outcomes, but it suffers from the lack of explaining the mechanistic basis. Could authors include the discussion on how HVEM might influence disease progression or immune response?
2. Please also discuss potential biases in patient cohort selection and how that could influence the results (the cohort counted 57 patients, so they selection criteria should be better presented and discussed).
3. IL-6 and IL-10 levels are discussed, by the lack of significant correlation with HVEM raises questions. Please acknowledge and discuss this.
4. Figures could be better annoyed and present in a higher quality (especially Figure 4). Please also consider improved labeling and captions for figures so they clearly explain the key observations and findings.
Author Response
Reviewer 1:
The manuscript analyzes the prognostic role of HVEM expression on monocytes and their various subsets in pancreatic ductal adenocarcinoma, PDAC. Authors demonstrate its correlation with reduced survival, aggressive clinicopathological features, and significant predictive value compared to CA19-9 as a biomarker. The study employs flow cytometry and survival analysis, highlighting the potential of HVEM as a novel biomarker for early detection and improved prognosis in PDAC.
In terms of originality and novelty, the research provides a novel perspective by focusing on HVEM expression in PDAC, which is less explored compared to other cancers. In addition its innovative approach to comparing HVEM to CA19-9 enhances the overall significance. Scientifically, the study is sound and easy to follow, with robust methodologies and detailed statistical analyses. Overall, this manuscript contributes to the field of cancer biomarker research and offers (potential) translational applications for PDAC prognosis and screening. Some minor comments below:
We thank the reviewer for the appreciation.
- The study demonstrates association between HVEM expression and survival outcomes, but it suffers from the lack of explaining the mechanistic basis. Could authors include in the discussion on how HVEM might influence disease progression or immune response? – We thank the reviewer for this interesting suggestion. We have included a part that focusses on the possible mechanism, how HVEM influences the tumor progression.
- Please also discuss potential biases in patient cohort selection and how that could influence the results (the cohort counted 57 patients, so they selection criteria should be better presented and discussed). – We thank the reviewer for this question. The inclusion and exclusion criteria of patient selection are below. We have included them in the material and method section too. PDAC patients: age over 40 years old, both gender, pathological confirmation of PDAC, no current cancer diagnosis.
Control patients: over 40 years old, both gender, no current cancer diagnosis, Patients who underwent surgical intervention due to chronic inflammatory condition in the gastrointestinal system (as mentioned in Table 2). Biases: known and unknown comorbidities of the patients, medical treatment of the comorbidities, Healthier patients may be more willing to participate in the study, Patients treated in highly specialized cancer centers may have different clinical characteristics than those treated in smaller clinics or general practices,
- IL-6 and IL-10 levels are discussed, by the lack of significant correlation with HVEM raises questions. Please acknowledge and discuss this. – We thank the reviewer for raising this point. We included IL-6 and IL-10, as these both cytokines are known to drive immune evasion during pancreatic cancer. As no one has shown any correlation between HVEM, IL-6 and IL-10, we wanted to explore if our marker of interested correlated with IL-6 and IL-10. However, it showed no correlation between them. We included a figure showing the functional role of monocytes during pancreatic cancer. In the coculture system too we did not observe any difference in the levels of IL-6 and IL-10 (Figure 4).
- Figures could be better annotated and present in a higher quality (especially Figure 4). Please also consider improved labeling and captions for figures so they clearly explain the key observations and findings. – We thank the reviewer for this suggestion. We improved the quality, labelling and captions of the figures.
Reviewer 2 Report
Comments and Suggestions for Authors
The manuscript titled "Increased HVEM expression on circulating monocytes and its subsets predicts reduced outcome in Pancreatic Ductal Adenocarcinoma patients" presents an interesting and clinically relevant study on the prognostic potential of HVEM in PDAC. The study is well-structured and provides valuable insights into the role of HVEM expression in monocytes and its subsets as potential biomarkers. However, there are several areas that require clarification, improvement, and further discussion. Below are specific comments regarding the manuscript's strengths and weaknesses.
Major Comments:
o The study includes 57 PDAC patients and 17 clinical controls. Given the variability in immune markers, how was the sample size determined?
o A power calculation should be provided to justify the robustness of the statistical findings.
o The manuscript demonstrates an association between HVEM expression and poor prognosis in PDAC. However, the biological mechanism underlying this association remains underexplored.
o It would strengthen the study to include a discussion on whether HVEM expression is a driver of aggressive tumor phenotypes or merely a bystander effect.
o While HVEM expression is compared with CA19-9, it would be beneficial to discuss how it compares with other emerging biomarkers for PDAC.
o Are there any synergistic effects between HVEM and other immune-related biomarkers?
o The study measures IL-6 and IL-10 levels but does not find a significant correlation with HVEM expression.
o Can the authors comment on potential reasons for this lack of correlation?
o The study highlights HVEM expression in different monocyte subsets. However, functional assays (e.g., cytokine production, phagocytic activity) are lacking.
o Have the authors considered performing single-cell RNA sequencing or functional validation to confirm whether these monocytes have distinct roles in PDAC progression?
Minor Comments:
· The introduction provides a thorough background but should briefly mention other immune checkpoint molecules in PDAC for broader context.
· Consider referencing more recent studies on tumor-associated monocytes/macrophages in PDAC.
· Were fluorescence minus one (FMO) controls used for HVEM detection?
· Provide more details on the ELISA methods, including the detection limits.
· Figure legends should clearly define abbreviations for a broader audience.
· In survival analysis, confidence intervals for hazard ratios should be reported.
· The discussion lacks a critical evaluation of study limitations. The authors should address potential biases, including inter-patient variability in immune responses.
· Would the findings be applicable to early-stage PDAC, or are they more relevant for advanced disease?
· The conclusion should emphasize how HVEM could be integrated into clinical practice.
· Are there any therapeutic implications for targeting HVEM in PDAC?
Author Response
Reviewer 2
Major Comments:
- The study includes 57 PDAC patients and 17 clinical controls. Given the variability in immune markers, how was the sample size determined?
We thank the reviewer for this question. This study was conducted as a prospective observational study with collection of the liquid and solid samples from PDAC patients and clinical controls followed by a retrospective analysis of the clinico-pathological data and correlation with several immunological markers in the liquid biopsies. Therefore, an upfront calculation of the samples sizes was not performed. We are planning to conduct another follow up study - including samples size calculation - based on the results of this current analysis. However, this will take another 2 years until results will be ready.
- A power calculation should be provided to justify the robustness of the statistical findings. – We thank the reviewer for this question. We used SPSS to calculate the posthoc power for monocytes, CM, IMM and NCM (PDAC vs. Control; independent t-test with welch correction). However, I have also read that post hoc power analyses are not useful, so I am not sure if it´s useful her, additionally the values are extremely high… Papers https://doi.org/10.1136/gpsych-2019-100069 https://doi.org/10.1002/gepi.22464
Power (Monocytes)=0,996 Power(CM)=0,996 Power(IMM)=0,984 Power(NCM)=0,998
- The manuscript demonstrates an association between HVEM expression and poor prognosis in PDAC. However, the biological mechanism underlying this association remains underexplored. – We are grateful to the reviewer for highlighting this aspect. In the revised manuscript, we investigated the immunosuppressive function of HVEM, as well as elucidated its regulatory influence on monocyte functionality by modulating phagocytic activity and cytokine production. Our observations revealed a tendency for an inverse correlation between HVEM-expressing monocytes and its subsets and the frequencies of Granzyme B in cytotoxic T cells. Furthermore, we observed that monocytes, upon interacting with tumor cells, undergo a loss of phagocytic capacity, concurrent with an increase in HVEM expression on their surface.
- It would strengthen the study to include a discussion on whether HVEM expression is a driver of aggressive tumor phenotypes or merely a bystander effect. – We thank the reviewer for this intriguing suggestion. We did discuss about HVEM being a driver of aggressive tumor phenotype.
- While HVEM expression is compared with CA19-9, it would be beneficial to discuss how it compares with other emerging biomarkers for PDAC. – We thank the reviewer for this nice suggestion. We have now included a supplementary figure showing the expression of HVEM to other markers like CRP, Bilirubin, CEA.
- Are there any synergistic effects between HVEM and other immune-related biomarkers? – we thank the reviewer for this point. We have discussed about the synergistic effects between HVEM and PD-1 in the discussion section.
- The study measures IL-6 and IL-10 levels but does not find a significant correlation with HVEM expression.Can the authors comment on potential reasons for this lack of correlation? – We would like to thank the reviewer for highlighting this issue. We incorporated IL-6 and IL-10 into our study due to their established role in promoting immune evasion during pancreatic cancer. Given the absence of documented correlation between HVEM, IL-6, and IL-10, we sought to investigate the potential relationship between our selected marker of interest and these cytokines. However, our findings revealed no discernible correlation between them. To this end, we measured the levels of IL-6 and IL-10 in a coculture system comprising monocytes from healthy donors and a human pancreatic cancer cell line. However, our analysis revealed no significant differences in the levels of IL-6 and IL-10 between the two groups.
- The study highlights HVEM expression in different monocyte subsets. However, functional assays (e.g., cytokine production, phagocytic activity) are lacking. We thank the reviewer for pointing this out. We performed functional assays, and the data is included in the revised manuscript version as Figure 4. We determined phagocytosis by analyzing the expression of CD206 and CD163 on different subsets of monocyte. Subsequently, we measured the cytokine level in the supernatant of the coculture.
- Have the authors considered performing single-cell RNA sequencing or functional validation to confirm whether these monocytes have distinct roles in PDAC progression? – We thank the reviewer for this interesting suggestion. We plan to do the same in our forthcoming projects.
Minor Comments:
- The introduction provides a thorough background but should briefly mention other immune checkpoint molecules in PDAC for broader context. – We thank the reviewer for this interesting suggestion. Now we have briefly mentioned about the checkpoint molecules in the introduction section.
- Consider referencing more recent studies on tumor-associated monocytes/macrophages in PDAC. – We thank the reviewer for this suggestion. We have included the recent studies on TAMs in PDAC now.
- Were fluorescence minus one (FMO) controls used for HVEM detection? – We thank the reviewer for raising this point. Yes, we did use FMO control to determine HVEM expression.
- Provide more details on the ELISA methods, including the detection limits. – We thank the reviewer for this advice. We have detailed the ELISA method with the detection limits.
- Figure legends should clearly define abbreviations for a broader audience. – we thank the reviewer for this suggestion. We have defined the abbreviation part clearly now.
- In survival analysis, confidence intervals for hazard ratios should be reported. We thank the reviewer for this point. We have now included the confidence interval and hazard ratios are reported in the results sections.
Monocytes (high:low): HR=6.103 and 95%CI=(2.055;18.13)
CM (high:low): HR=6.103 and 95%CI=(2.055;18.13)
IMM (high:low): HR=6.103 and 95%CI=(2.055;18.13)
NCM (high:low): HR=14.01 and 95%CI=(4.709;41.69)
- The discussion lacks a critical evaluation of study limitations. The authors should address potential biases, including inter-patient variability in immune responses. – We thank the reviewer for this suggestion. We have included points focusing on the critical evaluation of the limitation of this study.
- Would the findings be applicable to early-stage PDAC, or are they more relevant for advanced disease? – We thank the reviewer for this question. Our study mainly focused on preoperative sample
- The conclusion should emphasize how HVEM could be integrated into clinical practice. – We thank the reviewer for this comment. Now we have emphasized how HVEM could be integrated into clinical practice to be a prognostic tool in the conclusion section.
- Are there any therapeutic implications for targeting HVEM in PDAC? – We would like to thank the reviewer for highlighting this intriguing issue. The interaction of HVEMs with its inhibitory receptors has been demonstrated to induce immunosuppression in T cells, which in turn contributes to the progression of tumors in various solid cancers. However, the intricacies of HVEMs' interaction with its receptors during PDAC remain to be elucidated, underscoring the necessity for further research to ascertain its therapeutic potential.
Round 2
Reviewer 2 Report
Comments and Suggestions for Authors
The authors have taken into consideration all our previous comments; the article can be considered for publication.